# The Forward Order Law for Least Square *g*-Inverse of Multiple Matrix Products

**Zhiping Xiong *** 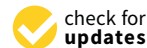 **and Zhongshan Liu**

School of Mathematics and Computational Science, Wuyi University, Jiangmen 529020, China; zsliu@163.com
* Correspondence: xzpwhere@163.com

**Abstract:** The generalized inverse has many important applications in the aspects of the theoretic research of matrices and statistics. One of the core problems of the generalized inverse is finding the necessary and sufficient conditions of the forward order laws for the generalized inverse of the matrix product. In this paper, by using the extremal ranks of the generalized Schur complement, we obtain some necessary and sufficient conditions for the forward order laws $A_1\{1,3\}A_2\{1,3\}\cdots A_n\{1,3\} \subseteq (A_1A_2\cdots A_n)\{1,3\}$ and $A_1\{1,4\}A_2\{1,4\}\cdots A_n\{1,4\} \subseteq (A_1A_2\cdots A_n)\{1,4\}$.

**Keywords:** forward order law; generalized inverse; maximal rank; matrix product; generalized Schur complement

## 1. Introduction

Throughout this paper, $C^{m\times n}$ denotes the set of $m \times n$ matrices with complex entries and $C^m$ denotes the set of $m$-dimensional vectors. $I_m$ denotes the identity matrix of order $m$, and $O_{m\times n}$ is the $m \times n$ matrix of all zero entries (if no confusion occurs, we will drop the subscript). For a matrix $A \in C^{m\times n}$, $A^*$, $r(A)$, $R(A)$, and $N(A)$ denote the conjugate transpose, the rank, the range space, and the null space of $A$, respectively. Furthermore, for the sake of simplicity in the later discussion, we will adopt the following notation for the matrix products with $A_i \in C^{m\times m}$, $i = 0, 1, 2, \cdots n - 1$,

$$\mathcal{A}_i = A_n A_{n-1} \cdots A_{n-i}. \tag{1}$$

Let $A \in C^{m\times n}$, a generalized inverse $X \in C^{n\times m}$ of $A$, be a matrix that satisfies some of the following four Penrose equations [1]:

$$(1)\ AXA = A, \ (2)\ XAX = X, \ (3)\ (AX)^* = AX, \ (4)\ (XA)^* = XA. \tag{2}$$

For a subset $\eta \subseteq \{1, 2, 3, 4\}$, the set of $n \times m$ matrices satisfying the equations that is contained in $\eta$ is denoted by $A\{\eta\}$. A matrix from $A\{\eta\}$ is called an $\{\eta\}$-inverse of $A$ and is denoted by $A^{(\eta)}$. For example, an $n \times m$ matrix $X$ of the set $A\{1\}$ is called a $\{1\}$-inverse of $A$ and is denoted by $X = A^{(1)}$. One usually denotes any $\{1,3\}$-inverse of $A$ as $A^{(1,3)}$, which is also called a least squares *g*-inverse of $A$. Any $\{1,4\}$-inverse of the set $A\{1,4\}$ is denoted by $A^{(1,4)}$, which is also called a minimum norm *g*-inverse of $A$. The unique $\{1,2,3,4\}$-inverse of $A$ is denoted by $A^\dagger$, which is also called the Moore–Penrose inverse of $A$. We refer the reader to [2,3] for basic results on the generalized inverses.

Theory and computations of the reverse (forward) order laws for generalized inverse are important in many branches of applied sciences, such as non-linear control theory [4], matrix analysis [2,5], statistics [4,6], and numerical linear algebra; see [3,6]. Suppose $A_i \in C^{m\times m}$, $i = 1, 2, \cdots, n$, and $b \in C^m$; the least squares technique (LS):

$$\min_{x\in C^m} \|(A_1 A_2 \cdots A_n)x - b\|_2$$

is used in many practical scientific problems [2,4,6,7]. Any solution $x$ of the above LS problem can be expressed as $x = (A_1 A_2 \cdots A_n)^{(1,3)} b$. If the LS problem is consistent, the minimum norm solution $x$ has the form $x = (A_1 A_2 \cdots A_n)^{(1,4)} b$. The unique minimal norm least squares solution $x$ of the LS problem is $x = (A_1 A_2 \cdots A_n)^{\dagger} b$. One problem concerning the above LS problem is under what conditions do the following reverse order laws hold?

$$A_n^{(i,j,\cdots,k)} A_{n-1}^{(i,j,\cdots,k)} \cdots A_1^{(i,j,\cdots,k)} = (A_1 A_2 \cdots A_n)^{(i,j,\cdots,k)}. \tag{3}$$

Another problem is under what conditions do the following forward order laws holds?

$$A_1^{(i,j,\cdots,k)} A_2^{(i,j,\cdots,k)} \cdots A_n^{(i,j,\cdots,k)} = (A_1 A_2 \cdots A_n)^{(i,j,\cdots,k)}, \tag{4}$$

where $(i, j, \cdots, k) \subseteq \eta \subseteq \{1, 2, 3, 4\}$.

The reverse order law for the generalized inverse of multiple matrix products yields a class of interesting problems that are fundamental in the theory of the generalized inverse of matrices; see [2,3,5,8,9]. As one of the core problems in reverse order laws, the necessary and sufficient conditions for the reverse order laws for the generalized inverse of matrix product hold, is useful in both theoretical study and practical scientific computing, this has attracted considerable attention, and many interesting results have been obtained; see [8,10–18].

The forward order law for generalized inverse of multiple matrix products (4) originally arose in studying the inverse of multiple matrix Kronecker products. Let $A_i$, $i = 1, 2, \cdots, n$, be $n$ nonsingular matrices, then the Kronecker product $A_1 \otimes A_2 \otimes \cdots \otimes A_n$ is nonsingular as well, and the inverse of $A_1 \otimes A_2 \otimes \cdots \otimes A_n$ satisfies the forward order law $A_1^{-1} \otimes A_2^{-1} \otimes \cdots \otimes A_n^{-1} = (A_1 \otimes A_2 \otimes \cdots \otimes A_n)^{-1}$. However, this so-called forward order law is not necessarily true for the matrix product, that is $A_1^{-1} A_2^{-1} \cdots A_n^{-1} \neq (A_1 A_2 \cdots A_n)^{-1}$. An interesting problem is, for given $\{i, j, \cdots, k\} \subseteq \eta$ and matrices $A_i$, $i = 1, 2, \cdots, n$, with $A_1 A_2 \cdots A_n$ meaningful, when $A_1^{(i,j,\cdots,k)} A_2^{(i,j,\cdots,k)} \cdots A_n^{(i,j,\cdots,k)} = (A_1 A_2 \cdots A_n)^{(i,j,\cdots,k)}$ or when

$$A_1 \{i, j, \cdots, k\} A_2 \{i, j, \cdots, k\} \cdots A_n \{i, j, \cdots, k\} = (A_1 A_2 \cdots A_n) \{i, j, \cdots, k\}$$

Recently, Z. Liu and Z. Xiong [19,20] studied the forward order law for $\{1,3\}$-inverses of three matrix products by using the maximal rank of the generalized Schur complement [21], and some necessary and sufficient conditions for $A_1 \{1,3\} A_2 \{1,3\} A_3 \{1,3\} \subseteq (A_1 A_2 A_3) \{1,3\}$ were derived. In this paper, we further study this subject, and some necessary and sufficient conditions by the ranks, the ranges, or the null spaces of the known matrices are provided for the following forward order laws:

$$A_1 \{1,3\} A_2 \{1,3\} \cdots A_n \{1,3\} \subseteq (A_1 A_2 \cdots A_n) \{1,3\} \tag{5}$$

and:

$$A_1 \{1,4\} A_2 \{1,4\} \cdots A_n \{1,4\} \subseteq (A_1 A_2 \cdots A_n) \{1,4\}. \tag{6}$$

The main tools of the later discussion are the following lemmas.

**Lemma 1** ([21]). *Let $A \in C^{m \times n}$, $B \in C^{m \times k}$, $C \in C^{l \times n}$, and $D \in C^{l \times k}$. Then:*

$$\max_{A^{(1,3)}} r(D - C A^{(1,3)} B) = \min \left\{ r \begin{pmatrix} A^* A & A^* B \\ C & D \end{pmatrix} - r(A), \ r \begin{pmatrix} B \\ D \end{pmatrix} \right\}.$$

**Lemma 2** ([5]). *Let $A \in \mathbb{C}^{m \times n}$, $B \in \mathbb{C}^{m \times k}$, and $C \in \mathbb{C}^{p \times n}$, then:*

$$(1) \quad r \begin{pmatrix} A, & B \end{pmatrix} = r(A) + r(E_A B) = r(B) + r(E_B A),$$

$$(2) \ r \begin{pmatrix} A \\ C \end{pmatrix} = r(A) + r(CF_A) = r(C) + r(AF_C),$$

*where the projectors $E_A = I - AA^\dagger$, $E_B = I - BB^\dagger$, $F_A = I - A^\dagger A$, $F_C = I - C^\dagger C$.*

**Lemma 3** ([2,3]). *Let $L$, $M$ be a complementary subspace of $C^n$, and let $P_{L,M}$ be the projector on $L$ along $M$. Then:*

$$(1) \ P_{L,M}A = A \Longleftrightarrow R(A) \subseteq L,$$

$$(2) \ AP_{L,M} = A \Longleftrightarrow N(A) \subseteq M.$$

**Lemma 4** ([2,3]). *Let $A \in C^{m \times n}$ and $X \in C^{n \times m}$, then:*

$$(1) \ X \in A\{1,3\} \Longleftrightarrow A^*AX = A'$$

$$(2) \ X \in A\{1,4\} \Longleftrightarrow XAA^* = A^*.$$

## 2. Main Results

In this section, we will present some necessary and sufficient conditions for the forward order law (5) by using the maximal ranks of some generalized Schur complement forms. Let:

$$
\begin{aligned}
S_{(A_1 A_2 \cdots A_n)^*} &= S_{\mu^*} = (A_1 A_2 \cdots A_n)^* - (A_1 A_2 \cdots A_n)^* A_1 A_2 \cdots A_n X_1 X_2 \cdots X_n \\
&= \mu^* - \mu^* \mu X_1 X_2 \cdots X_n,
\end{aligned}
\tag{7}
$$

*where $A_i \in C^{m \times m}$, $X_i \in A_i\{1,3\}$, $i = 1,2,\cdots,n$, and $\mu = A_1 A_2 \cdots A_n$.*

For the convenience of the readers, we first give a brief outline of the basic idea. From the formula (1) in Lemma 4, we know that the forward order law (5) holds if and only if

$$\mu^* \mu X_1 X_2 \cdots X_n = \mu^*$$

holds for any $X_i \in A_i\{1,3\}$, $i = 1,2,\cdots,n$, and $\mu = A_1 A_2 \cdots A_n$, which is equivalent to the following identity:

$$\max_{X_1, X_2, \cdots, X_n} r(S_{(A_1 A_2 \cdots A_n)^*}) = \max_{X_1, X_2, \cdots, X_n} r(S_{\mu^*}) = 0. \tag{8}$$

Hence, we can present the equivalent conditions for the forward order law (5) if the concrete expression of the maximal rank involved in the identity (8) is derived. The relative results are included in the following lemma.

**Lemma 5.** *Let $A_i \in C^{m \times m}$, $X_i \in A_i\{1,3\}$, $i = 1,2,\cdots,n$. Let $\mathcal{A}_i$ be as in (1) and $S_{\mu^*}$ be as in (7), and $E_{A_i} = I - A_i A_i^\dagger$, $i = 1,2,\cdots,n$ be $n$ projectors. Then:*

$$
\begin{aligned}
&\max_{X_1, X_2, \cdots, X_n} r(S_{(A_1 A_2 \cdots A_n)^*}) = \max_{X_1, X_2, \cdots, X_n} r(S_{\mu^*}) \\
&= r \left( \mu^* \mu - \mu^* \mathcal{A}_{n-1}, \quad \mu^* \mathcal{A}_{n-2} E_{A_1}, \quad \mu^* \mathcal{A}_{n-3} E_{A_2}, \quad \cdots, \quad \mu^* \mathcal{A}_0 E_{A_{n-1}}, \quad \mu^* E_{A_n} \right) \\
&= r \begin{pmatrix} \mu^* \mu - \mu^* \mathcal{A}_{n-1} & \mu^* \mathcal{A}_{n-2} & \mu^* \mathcal{A}_{n-3} & \cdots & \mu^* \mathcal{A}_0 & \mu^* \\ O & A_1^* & O & \cdots & O & O \\ O & O & A_2^* & \cdots & O & O \\ \vdots & \vdots & \vdots & \ddots & \vdots & \vdots \\ O & O & O & \cdots & A_{n-1}^* & O \\ O & O & O & \cdots & O & A_n^* \end{pmatrix} - \sum_{i=1}^{n} r(A_i).
\end{aligned}
$$

**Proof.** By Lemma 1 and the formula (2) of Lemma 2, we have:

$$
\begin{aligned}
\max_{X_n} r(S_{(A_1 A_2 \cdots A_n)^*}) &= \max_{X_n} r(S_{\mu^*}) \\
&= \max_{X_n} r(\mu^* - \mu^* \mu X_1 X_2 \cdots X_n) \\
&= \min \left\{ r \begin{pmatrix} A_n^* A_n & A_n^* \\ \mu^* \mu X_1 X_2 \cdots X_{n-1} & \mu^* \end{pmatrix} - r(A_n), \ r \begin{pmatrix} I_m \\ \mu^* \end{pmatrix} \right\} \\
&= \min \left\{ r \begin{pmatrix} O & A_n^* \\ \mu^* \mu X_1 X_2 \cdots X_{n-1} - \mu^* A_n & \mu^* \end{pmatrix} - r(A_n), \ m \right\} \\
&= r \left[ \left( \mu^* \mu X_1 X_2 \cdots X_{n-1} - \mu^* A_n, \ \mu^* \right) \cdot F_{(O, \ A_n^*)} \right] \\
&= r \left( \mu^* \mu X_1 X_2 \cdots X_{n-1} - \mu^* A_n, \ \mu^* E_{A_n} \right) \\
&= r \left[ \left( \mu^* A_n, \ \mu^* E_{A_n} \right) - \mu^* \mu X_1 X_2 \cdots X_{n-1} \left( I_m, \ O \right) \right].
\end{aligned} \tag{9}
$$

According to Lemma 1, the formula (2) in Lemma 2, and Equations (1) and (9), we have:

$$
\begin{aligned}
\max_{X_{n-1}, X_n} r(S_{(A_1 A_2 \cdots A_n)^*}) &= \max_{X_{n-1}, X_n} r(S_{\mu^*}) \\
&= \max_{X_{n-1}} r \left[ \left( \mu^* A_n, \ \mu^* E_{A_n} \right) - \mu^* \mu X_1 X_2 \cdots X_{n-1} \left( I_m, \ O \right) \right] \\
&= \min \left\{ r \begin{pmatrix} A_{n-1}^* A_{n-1} & A_{n-1}^* & O \\ \mu^* \mu X_1 X_2 \cdots X_{n-2} & \mu^* A_n & \mu^* E_{A_n} \end{pmatrix} - r(A_{n-1}), \ r \begin{pmatrix} I_m & O \\ \mu^* A_n & \mu^* E_{A_n} \end{pmatrix} \right\} \\
&= \min \left\{ r \begin{pmatrix} O & A_{n-1}^* & O \\ \mu^* \mu X_1 \cdots X_{n-2} - \mu^* \mathcal{A}_1 & \mu^* A_n & \mu^* E_{A_n} \end{pmatrix} - r(A_{n-1}), \ m + r(\mu^* E_{A_n}) \right\} \tag{10} \\
&= \min \left\{ r \left[ \left( \mu^* \mu X_1 X_2 \cdots X_{n-2} - \mu^* \mathcal{A}_1, \ \mu^* A_n, \ \mu^* E_{A_n} \right) F_{(O, \ A_{n-1}^*, \ O)} \right], \right. \\
&\qquad \left. m + r(\mu^* E_{A_n}) \right\} \\
&= r \left( \mu^* \mu X_1 X_2 \cdots X_{n-2} - \mu^* \mathcal{A}_1, \ \mu^* \mathcal{A}_0 E_{A_{n-1}}, \ \mu^* E_{A_n} \right) \\
&= r \left[ \left( \mu^* \mathcal{A}_1, \ \mu^* \mathcal{A}_0 E_{A_{n-1}}, \ \mu^* E_{A_n} \right) - \mu^* \mu X_1 X_2 \cdots X_{n-2} \left( I_m, \ O, \ O \right) \right].
\end{aligned}
$$

Again, by Lemma 1, the formula (2) in Lemma 2 and the results in (1) and (10), we have:

$$
\max_{X_{n-2},X_{n-1},X_n} r(S_{(A_1A_2\cdots A_n)^*}) = \max_{X_{n-2},X_{n-1},X_n} r(S_{\mu^*})
$$

$$
= \max_{X_{n-2}} r\left[\left(\mu^*\mathcal{A}_1,\ \mu^*\mathcal{A}_0 E_{A_{n-1}},\ \mu^* E_{A_n}\right) - \mu^*\mu X_1 X_2\cdots X_{n-2}\left(I_m,\ O,\ O\right)\right]
$$

$$
= \min\left\{ r\begin{pmatrix} A_{n-2}^* A_{n-2} & A_{n-2}^* & O & O \\ \mu^*\mu X_1 X_2\cdots X_{n-3} & \mu^*\mathcal{A}_1 & \mu^*\mathcal{A}_0 E_{A_{n-1}} & \mu^* E_{A_n} \end{pmatrix} - r(A_{n-2}), \right.
$$

$$
\left. r\begin{pmatrix} I_m & O & O \\ \mu^*\mathcal{A}_1 & \mu^*\mathcal{A}_0 E_{A_{n-1}} & \mu^* E_{A_n} \end{pmatrix} \right\}
$$

$$
= \min\left\{ r\begin{pmatrix} O & A_{n-2}^* & O & O \\ \mu^*\mu X_1 X_2\cdots X_{n-3} - \mu^*\mathcal{A}_2 & \mu^*\mathcal{A}_1 & \mu^*\mathcal{A}_0 E_{A_{n-1}} & \mu^* E_{A_n} \end{pmatrix} - r(A_{n-2}), \right. \tag{11}
$$

$$
\left. m + r\left(\mu^*\mathcal{A}_0 E_{A_{n-1}},\ \mu^* E_{A_n}\right) \right\}
$$

$$
= \min\left\{ r\left[\left(\mu^*\mu X_1\cdots X_{n-3} - \mu^*\mathcal{A}_2,\ \mu^*\mathcal{A}_1,\ \mu^*\mathcal{A}_0 E_{A_{n-1}},\ \mu^* E_{A_n}\right) F_{\left(O,\ A_{n-2}^*,\ O,\ O\right)}\right], \right.
$$

$$
\left. m + r\left(\mu^*\mathcal{A}_0 E_{A_{n-1}},\ \mu^* E_{A_n}\right) \right\}
$$

$$
= r\left(\mu^*\mu X_1 X_2\cdots X_{n-3} - \mu^*\mathcal{A}_2,\ \mu^*\mathcal{A}_1 E_{A_{n-2}},\ \mu^*\mathcal{A}_0 E_{A_{n-1}},\ \mu^* E_{A_n}\right)
$$

$$
= r\left[\left(\mu^*\mathcal{A}_2,\ \mu^*\mathcal{A}_1 E_{A_{n-2}},\ \mu^*\mathcal{A}_0 E_{A_{n-1}},\ \mu^* E_{A_n}\right) - \mu^*\mu X_1\cdots X_{n-3}\left(I_m,\ O,\ O,\ O\right)\right].
$$

Suppose $X_0 = I_m$. We contend that, for $2 \le i \le n-1$,

$$
\max_{X_{n-i},X_{n-i+1},\cdots,X_n} r(S_{(A_1A_2\cdots A_n)^*}) = \max_{X_{n-i},X_{n-i+1},\cdots,X_n} r(S_{\mu^*})
$$

$$
= \max_{X_{n-i}} r\left[\left(\mu^*\mathcal{A}_{i-1},\ \mu^*\mathcal{A}_{i-2} E_{A_{n-i+1}},\ \cdots,\ \mu^*\mathcal{A}_0 E_{A_{n-1}},\ \mu^* E_{A_n}\right)\right.
$$

$$
\left. - \mu^*\mu X_1 X_2\cdots X_{n-i}\left(I_m,\ O,\ \cdots,\ O,\ O\right)\right] \tag{12}
$$

$$
= r\left[\left(\mu^*\mathcal{A}_i,\ \mu^*\mathcal{A}_{i-1} E_{A_{n-i}},\ \mu^*\mathcal{A}_{i-2} E_{A_{n-i+1}},\ \cdots,\ \mu^*\mathcal{A}_0 E_{A_{n-1}},\ \mu^* E_{A_n}\right)\right.
$$

$$
\left. - \mu^*\mu X_1 X_2\cdots X_{n-i-1}\left(I_m,\ O,\ O,\ \cdots,\ O,\ O\right)\right].
$$

We proceed by induction on $i$. For $i = 2$, from (11), the equality relation (12) has been proven. Assuming that (12) is true for $i-1$ ($i \ge 3$), that is:

$$
\max_{X_{n-i},X_{n-i+1},\cdots,X_n} r(S_{(A_1A_2\cdots A_n)^*}) = \max_{X_{n-i+1},X_{n-i+2},\cdots,X_n} r(S_{\mu^*})
$$

$$
= \max_{X_{n-i+1}} r\left[\left(\mu^*\mathcal{A}_{i-2} E_{A_{n-i+1}},\ \mu^*\mathcal{A}_{i-3} E_{A_{n-i+2}},\ \cdots,\ \mu^*\mathcal{A}_0 E_{A_{n-1}},\ \mu^* E_{A_n}\right)\right. \tag{13}
$$

$$
\left. - \mu^*\mu X_1 X_2\cdots X_{n-i+1}\left(I_m,\ O,\ \cdots,\ O,\ O\right)\right]
$$

$$
= r\left[\left(\mu^*\mathcal{A}_{i-1},\ \mu^*\mathcal{A}_{i-2} E_{A_{n-i+1}},\ \cdots,\ \mu^*\mathcal{A}_0 E_{A_{n-1}},\ \mu^* E_{A_n}\right)\right.
$$

$$
\left. - \mu^*\mu X_1 X_2\cdots X_{n-i}\left(I_m,\ O,\ \cdots,\ O,\ O\right)\right].
$$

Next, we will prove that (12) is also true for $i$. In fact, by Lemma 1, the formula (2) in Lemma 2, and the results in (13) and (1), we have:

$$
\max_{X_{n-i},X_{n-i+1},\cdots,X_n} r(S_{(A_1A_2\cdots A_n)^*}) = \max_{X_{n-i},X_{n-i+1},\cdots,X_n} r(S_{\mu^*})
$$

$$
= \max_{X_{n-i}} r\left(\left(\mu^*\mathcal{A}_{i-1}, \quad \mu^*\mathcal{A}_{i-2}E_{A_{n-i+1}}, \quad \cdots, \quad \mu^*\mathcal{A}_0E_{A_{n-1}}, \quad \mu^*E_{A_n}\right)\right.
$$

$$
\left. - \mu^*\mu X_1X_2\cdots X_{n-i}\left(I_m, \quad O, \quad \cdots, \quad O, \quad O\right)\right)
$$

$$
= \min\left\{ r\left(\begin{matrix} A_{n-i}^*A_{n-i} & A_{n-i}^* & O & \cdots & O \\ \mu^*\mu X_1X_2\cdots X_{n-i-1} & \mu^*\mathcal{A}_{i-1} & \mu^*\mathcal{A}_{i-2}E_{A_{n-i+1}} & \cdots & \mu^*E_{A_n} \end{matrix}\right) - r(A_{n-i}), \right.
$$

$$
\left. r\left(\begin{matrix} I_m & O & \cdots & O \\ \mu^*\mathcal{A}_{i-1} & \mu^*\mathcal{A}_{i-2}E_{A_{n-i+1}} & \cdots & \mu^*E_{A_n} \end{matrix}\right) \right\} \tag{14}
$$

$$
= \min\left\{ r\left(\begin{matrix} O & A_{n-i}^* & O & \cdots & O \\ \mu^*\mu X_1X_2\cdots X_{n-i-1} - \mu^*\mathcal{A}_i & \mu^*\mathcal{A}_{i-1} & \mu^*\mathcal{A}_{i-2}E_{A_{n-i+1}} & \cdots & \mu^*E_{A_n} \end{matrix}\right) - r(A_{n-i}), \right.
$$

$$
\left. m + r\left(\mu^*\mathcal{A}_{i-2}E_{A_{n-i+1}}, \quad \cdots, \quad \mu^*E_{A_n}\right) \right\}
$$

$$
= r\left(\mu^*\mu X_1X_2\cdots X_{n-i-1} - \mu^*\mathcal{A}_i, \quad \mu^*\mathcal{A}_{i-1}E_{A_{n-i}}, \quad \mu^*\mathcal{A}_{i-2}E_{A_{n-i+1}}, \quad \cdots, \quad \mu^*E_{A_n}\right)
$$

$$
= r\left[\left(\mu^*\mathcal{A}_i, \quad \mu^*\mathcal{A}_{i-1}E_{A_{n-i}}, \quad \mu^*\mathcal{A}_{i-2}E_{A_{n-i+1}}, \quad \cdots, \quad \mu^*\mathcal{A}_0E_{A_{n-1}}, \quad \mu^*E_{A_n}\right)\right.
$$

$$
\left. - \mu^*\mu X_1X_2\cdots X_{n-i-1}\left(I_m, \quad O, \quad O, \quad \cdots, \quad O, \quad O\right)\right].
$$

That is, the equality relation (12) has been proven. Specifically, when $i = n - 1$, we have:

$$
\max_{X_{n-i},X_{n-i+1},\cdots,X_n} r(S_{(A_1A_2\cdots A_n)^*}) = \max_{X_1,X_2,\cdots,X_n} r(S_{\mu^*})
$$

$$
= \max_{X_1} r\left[\left(\mu^*\mathcal{A}_{n-2}, \quad \mu^*\mathcal{A}_{n-3}E_{A_2}, \quad \mu^*\mathcal{A}_{n-4}E_{A_3}, \quad \cdots, \quad \mu^*\mathcal{A}_0E_{A_{n-1}}, \quad \mu^*E_{A_n}\right)\right.
$$

$$
\left. - \mu^*\mu X_1\left(I_m, \quad O, \quad O, \quad \cdots, \quad O, \quad O\right)\right]
$$

$$
= r\left[\left(\mu^*\mathcal{A}_{n-1}, \quad \mu^*\mathcal{A}_{n-2}E_{A_1}, \quad \mu^*\mathcal{A}_{n-3}E_{A_2}, \quad \cdots, \quad \mu^*\mathcal{A}_0E_{A_{n-1}}, \quad \mu^*E_{A_n}\right)\right.
$$

$$
\left. - \mu^*\mu X_0\left(I_m, \quad O, \quad O, \quad \cdots, \quad O, \quad O\right)\right]
$$

$$
= r\left(\mu^*\mu - \mu^*\mathcal{A}_{n-1}, \quad \mu^*\mathcal{A}_{n-2}E_{A_1}, \quad \mu^*\mathcal{A}_{n-3}E_{A_2}, \quad \cdots, \quad \mu^*\mathcal{A}_0E_{A_{n-1}}, \quad \mu^*E_{A_n}\right). \tag{15}
$$

Combining (15) with Lemma 2, we finally have:

$$
\max_{X_{n-i},X_{n-i+1},\cdots,X_n} r(S_{(A_1A_2\cdots A_n)^*}) = \max_{X_1,X_2,\cdots,X_n} r(S_{\mu^*})
$$

$$
= r\left(\mu^*\mu - \mu^*\mathcal{A}_{n-1}, \quad \mu^*\mathcal{A}_{n-2}E_{A_1}, \quad \mu^*\mathcal{A}_{n-3}E_{A_2}, \quad \cdots, \quad \mu^*\mathcal{A}_0E_{A_{n-1}}, \quad \mu^*E_{A_n}\right)
$$

$$
= r\begin{pmatrix} \mu^*\mu - \mu^*\mathcal{A}_{n-1} & \mu^*\mathcal{A}_{n-2} & \mu^*\mathcal{A}_{n-3} & \cdots & \mu^*\mathcal{A}_0 & \mu^* \\ O & A_1^* & O & \cdots & O & O \\ O & O & A_2^* & \cdots & O & O \\ \vdots & \vdots & \vdots & \ddots & \vdots & \vdots \\ O & O & O & \cdots & A_{n-1}^* & O \\ O & O & O & \cdots & O & A_n^* \end{pmatrix} - \sum_{i=1}^{n} r(A_i).
$$

$\square$

From Lemma 5, Lemma 2, and Lemma 3, we immediately obtain the following theorem by Equation (8).

**Theorem 1.** *Let $A_i \in C^{m \times m}$, $i = 1, 2, \cdots, n$, $\mathcal{A}_i$ be as in (1) and $\mu = A_1 A_2 \cdots A_n$. Then, the following statements are equivalent:*

*(1)* $\quad A_1\{1,3\} A_2\{1,3\} \cdots A_n\{1,3\} \subseteq (A_1 A_2 \cdots A_n)\{1,3\};$

*(2)* $\quad r\left(\mu^* \mu - \mu^* \mathcal{A}_{n-1}, \quad \mu^* \mathcal{A}_{n-2} E_{A_1}, \quad \mu^* \mathcal{A}_{n-3} E_{A_2}, \quad \cdots, \quad \mu^* \mathcal{A}_0 E_{A_{n-1}}, \quad \mu^* E_{A_n}\right) = 0;$

*(3)* $\quad \mu^* \mu - \mu^* \mathcal{A}_{n-1} = O$ and $\mu^* \mathcal{A}_i E_{A_{n-i-1}} = O$ $(i = 0, 1, \cdots, n-2)$ and $\mu^* E_{A_n} = O;$

*(4)* $\quad \mu^* \mu = \mu^* \mathcal{A}_{n-1}$ and $N(\mu^* \mathcal{A}_i) \supseteq N(A^*_{n-i-1})$ $(i = 0, 1, \cdots, n-2)$ and $N(\mu^*) \supseteq N(A^*_n);$

*(5)*
$$r\begin{pmatrix} \mu^* \mu - \mu^* \mathcal{A}_{n-1} & \mu^* \mathcal{A}_{n-2} & \mu^* \mathcal{A}_{n-3} & \cdots & \mu^* \mathcal{A}_0 & \mu^* \\ O & A_1^* & O & \cdots & O & O \\ O & O & A_2^* & \cdots & O & O \\ \vdots & \vdots & \vdots & \ddots & \vdots & \vdots \\ O & O & O & \cdots & A_{n-1}^* & O \\ O & O & O & \cdots & O & A_n^* \end{pmatrix} = \sum_{i=1}^{n} r(A_i).$$

**Proof.** (1)$\Longleftrightarrow$(2). It is easy to see that the inclusion $X_1 X_2 \cdots X_n \in (A_1 A_2 \cdots A_n)\{1,3\}$ holds for any $X_i \in A_i\{1,3\}$, $i = 1, 2, \cdots, n$ if and only if:

$$\max_{X_1, X_2, \cdots, X_n} r((A_1 A_2 \cdots A_n)^* - (A_1 A_2 \cdots A_n)^* A_1 A_2 \cdots A_n X_1 X_2 \cdots X_n)$$
$$= \max_{X_1, X_2, \cdots, X_n} r(\mu^* - \mu^* \mu X_1 X_2 \cdots X_n) = 0.$$

From Lemma 5, we obtain (1)$\Longleftrightarrow$(2).

(2)$\Longleftrightarrow$(3). In fact, $r(A) = 0$ if and only if $A = O$, so (2)$\Longleftrightarrow$(3) is obvious.

(3)$\Longleftrightarrow$(4). From (3), we have $\mu^* \mu - \mu^* \mathcal{A}_{n-1} = O \Leftrightarrow \mu^* \mu = \mu^* \mathcal{A}_{n-1}$. On the other hand, by (3), we obtain:

$$\mu^* \mathcal{A}_{n-2} E_{A_1} = \mu^* \mathcal{A}_{n-2}(I_m - A_1 A_1^\dagger) = O.$$

That is,

$$\mu^* \mathcal{A}_{n-2} = \mu^* \mathcal{A}_{n-2} A_1 A_1^\dagger = \mu^* \mathcal{A}_{n-2} P_{R(A_1), N(A_1^*)}. \tag{16}$$

According to the formula (2) of Lemma 3, it is known that the formula of (13) is equivalent to:

$$N(\mu^* \mathcal{A}_{n-2}) \supseteq N(A_1^*).$$

Similarly, we can show that:

$$\mu^* \mathcal{A}_i E_{n-i-1} = O \ (i = 0, 1, \cdots, n-2)$$

is equivalent to:

$$N(\mu^* \mathcal{A}_i) \supseteq N(A_{n-i-1}^*) \ (i = 0, 1, \cdots, n-2).$$

Hence (3)$\Longleftrightarrow$(4).

(2)$\Longleftrightarrow$ (5). By Lemma 2, we have:

$$r\left(\mu^*\mu - \mu^*\mathcal{A}_{n-1},\ \mu^*\mathcal{A}_{n-2}E_{A_1},\ \mu^*\mathcal{A}_{n-3}E_{A_2},\ \cdots,\ \mu^*\mathcal{A}_0 E_{A_{n-1}},\ \mu^* E_{A_n}\right)$$

$$= r\left(\mu^*\mu - \mu^*\mathcal{A}_{n-1},\ \mu^*\mathcal{A}_{n-2}F_{A_1^*},\ \mu^*\mathcal{A}_{n-3}F_{A_2^*},\ \cdots,\ \mu^*\mathcal{A}_0 F_{A_{n-1}^*},\ \mu^* F_{A_n^*}\right)$$

$$= r\begin{pmatrix} \mu^*\mu - \mu^*\mathcal{A}_{n-1} & \mu^*\mathcal{A}_{n-2} & \mu^*\mathcal{A}_{n-3} & \cdots & \mu^*\mathcal{A}_0 & \mu^* \\ O & A_1^* & O & \cdots & O & O \\ O & O & A_2^* & \cdots & O & O \\ \vdots & \vdots & \vdots & \ddots & \vdots & \vdots \\ O & O & O & \cdots & A_{n-1}^* & O \\ O & O & O & \cdots & O & A_n^* \end{pmatrix} - \sum_{i=1}^{n} r(A_i). \tag{17}$$

From (17), we have:

$$r\left(\mu^*\mu - \mu^*\mathcal{A}_{n-1},\ \mu^*\mathcal{A}_{n-2}E_{A_1},\ \mu^*\mathcal{A}_{n-3}E_{A_2},\ \cdots,\ \mu^*\mathcal{A}_0 E_{A_{n-1}},\ \mu^* E_{A_n}\right) = 0,$$

if and only if:

$$r\begin{pmatrix} \mu^*\mu - \mu^*\mathcal{A}_{n-1} & \mu^*\mathcal{A}_{n-2} & \mu^*\mathcal{A}_{n-3} & \cdots & \mu^*\mathcal{A}_0 & \mu^* \\ O & A_1^* & O & \cdots & O & O \\ O & O & A_2^* & \cdots & O & O \\ \vdots & \vdots & \vdots & \ddots & \vdots & \vdots \\ O & O & O & \cdots & A_{n-1}^* & O \\ O & O & O & \cdots & O & A_n^* \end{pmatrix} = \sum_{i=1}^{n} r(A_i).$$

The proof of Theorem 1 is completed. $\square$

By Lemma 4, we know that $X \in A\{1,4\}$ if and only if $X^* \in A\{1,3\}$. Therefore, from the results obtained in Theorem 1, we can get the necessary and sufficient conditions for the forward order law (6), and hence provide the following theorem without the proof.

**Theorem 2.** *Let* $A_i \in C^{m\times m}$, $i = 1, 2, \cdots, n$, $\mu = A_1 A_2 \cdots A_n$, *and* $\mathbb{A}_i = A_i A_{i-1} \cdots A_1$, $i = 1, 2, \cdots, n$. *Then, the following statements are equivalent:*

(1)     $A_1\{1,4\}A_2\{1,4\}\cdots A_n\{1,4\} \subseteq (A_1 A_2 \cdots A_n)\{1,4\}$;

(2)     $r\begin{pmatrix} \mathbb{A}_n\mu^* - \mu\mu^* \\ F_{A_n}\mathbb{A}_{n-1}\mu^* \\ F_{A_{n-1}}\mathbb{A}_{n-2}\mu^* \\ \vdots \\ F_{A_2}\mathbb{A}_1\mu^* \\ F_{A_1}\mu^* \end{pmatrix} = 0$;

(3)     $\mathbb{A}_n\mu^* - \mu\mu^* = O$ *and* $F_{A_i}\mathbb{A}_{i-1}\mu^* = O$, $(i = 2, 3, \cdots n)$ *and* $F_{A_1}\mu^* = O$;

(4)     $\mathbb{A}_n\mu^* = \mu\mu^*$ *and* $R(\mathbb{A}_{i-1}\mu^*) \subseteq R(A_i^*)$, $(i = 2, 3, \cdots n)$ *and* $R(\mu^*) \subseteq R(A_1^*)$;

(5)     $r\begin{pmatrix} \mathbb{A}_n\mu^* - \mu\mu^* & O & O & \cdots & O & O \\ \mathbb{A}_{n-1}\mu^* & O & O & \cdots & O & A_n^* \\ \mathbb{A}_{n-2}\mu^* & O & O & \cdots & A_{n-1}^* & O \\ \vdots & \vdots & \vdots & & \vdots & \vdots \\ \mathbb{A}_1\mu^* & O & A_2^* & \cdots & O & O \\ \mu^* & A_1^* & O & \cdots & O & O \end{pmatrix} = \sum_{i=1}^{n} r(A_i).$

### 3. Conclusions

In this paper, we have studied the forward order laws for the $\{1, 3\}$-inverse and $\{1, 3\}$-inverse of a product of multiple matrices. By using the expressions for maximal ranks of the generalized Schur complement, we obtained some necessary and sufficient conditions for $A_1\{1, 3\} \cdots A_n\{1, 3\} \subseteq (A_1 A_2 \cdots A_n)\{1, 3\}$ and $A_1\{1, 4\} \cdots A_n\{1, 4\} \subseteq (A_1 A_2 \cdots A_n)\{1, 4\}$. In the near future, we will study more challenging problems to find out the important applications of forward order laws in many algorithms for the computation of the least squares technique (LS) of matrix equations $(A_1 A_2 \cdots A_n)x = b$.

**Author Contributions:** All authors have equally contributed to this work. All authors read and approved the final manuscript.

**Funding:** This work was supported by the project for characteristic innovation of 2018 Guangdong University and the National Science Foundation of China (No: 11771159, 11571004) and the Guangdong Natural Science Fund of China (Grant No: 2014A030313625).

**Acknowledgments:** The authors would like to thank the anonymous referees for their very detailed comments and constructive suggestions, which greatly improved the presentation of this paper.

**Conflicts of Interest:** The authors declare no conflict of interest.

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
