# Peer review of "The Forward Order Law for Least Squareg-Inverse of Multiple Matrix Products"

_mathematics, doi:10.3390/math7030277_

Round 1
Reviewer 1 Report
The results in this article are in general correct. the article can be accepted for publication. Some changes in the presentation are needed: A section with conclusions and future thoughts on the problem. The problem seem to have several applications, and there should be a clarification on this. Please check for typos.
Author Response
Revision Letter
Dear Sir,
We have received your letter dated 06-Mar-2019, giving us your advices for revision of the manuscript (mathematics-443440). We appreciate very much for your comments and the valuable suggestions. The manuscript has been carefully revised as the comments and the response listed as followed.
Reviewer 's comment (1): Some changes in the presentation are needed: A section with conclusions and future thoughts on the problem.
Reply: We have revised according to this suggestion in revised MS. See revised MS, page 9.
Reviewer's comment (2): The problem seems to have several applications, and there should be a clarification on this.
Reply: We have revised according to this suggestion in revised MS. See revised MS, page 2, lines 10-14.
The reviewer gave us many concrete suggestions to improve the manuscript. We thank them very much for the positive and constructive comments to the MS in the ACKNOWLEDGMENTS.
Thank you very much!
Sincerely yours
Dr. Zhiping Xiong
School of Mathematics and Computational Science, Wuyi University, Jiangmen
529020, P. R. China
Email: xzpwhere@163.com
Reviewer 2 Report
The present article deals with finding necessary and sufficient conditions for the forward order
laws for generalized inverse of matrix product. The article is well written, well structured and well documented. The main results are presented with clarity.
Comments
There are some minor english style mistakes to be corrected.
- In the sentence
"The reverse order law for generalized inverse of multiple matrix products yield a class of
interesting problems that are fundamental in the theory of generalized inverse of matrices" "yield" should be transformed to "yields".
- The sentence "As one of the core problems in reverse order laws, the necessary and sufficient
conditions for the reverse order laws for generalized inverse of matrix product holds, is useful
in both theoretical study and practical scientific computing, which has attracted considerable
attentions and many interesting results have been obtained,...", is not well built.
The authors should consider something like
"As one of the core problems in reverse order laws, the necessary and sufficient
conditions for the reverse order laws for generalized inverse of matrix product to hold, is useful
in both theoretical study and practical scientific computing. This has attracted considerable
attentions and many interesting results have been obtained,..."
Author Response
Revision Letter
Dear Sir,
We have received your letter dated 06-Mar-2019, giving us your advices for revision of the manuscript (mathematics-443440). We appreciate very much for your comments and the valuable suggestions. The manuscript has been carefully revised as the comments and the response listed as followed.
Reviewer 's comment (1): In the sentence "The reverse order law for generalized inverse of multiple matrix products yield a class of interesting problems that are fundamental in the theory of generalized inverse of matrices" "yield" should be transformed to "yields".
Reply: In revised MS, we have revised this error. See revised MS, page 2, lines 23.
Reviewer 's comment (2): The sentence "As one of the core problems in reverse order laws, the necessary and sufficient conditions for the reverse order laws for generalized inverse of matrix product holds, is useful in both theoretical study and practical scientific computing, which has attracted considerable attentions and many interesting results have been obtained,...", is not well built.
The authors should consider something like "As one of the core problems in reverse order laws, the necessary and sufficient conditions for the reverse order laws for generalized inverse of matrix product to hold, is useful in both theoretical study and practical scientific computing. This has attracted considerable attentions and many interesting results have been obtained,...".
Reply: We have revised according to this suggestion in revised MS. See revised MS, page 2, lines 25-28.
The reviewers gave us many concrete suggestions to improve the manuscript. We thank them very much for the positive and constructive comments to the MS in the ACKNOWLEDGMENTS.
Thank you very much!
Sincerely yours
Dr. Zhiping Xiong
School of Mathematics and Computational Science, Wuyi University, Jiangmen
529020, P. R. China
Email: xzpwhere@163.com